Seasonal dynamics of terrestrial vertebrate abundance between Amazonian flooded and unflooded forests

Costa Hugo C.M. hugocmcosta@gmail.com 1
Peres Carlos A. 2
Abrahams Mark I. 3
1 Programa de Pós-graduação em Zoologia, Museu Paraense Emílio Goeldi , Belém , Brazil
2 Centre for Ecology, Evolution and Conservation, School of Environmental Sciences, University of East Anglia , Norwich , United Kingdom
3 Field Conservation and Science Department, Bristol Zoological Society , Bristol , United Kingdom
Wilson Matthew
Electronic publication date: 2018 Jun 27
Publication date: 2018
Volume: 6
Electronic Location ID: e5058
Received 2018 Mar 10; Accepted 2018 Jun 2
Copyright: ©2018 Costa et al.
Copyright year: 2018
Copyright holder: Costa et al.
License: This is an open access article distributed under the terms of the Creative Commons Attribution License, which permits unrestricted use, distribution, reproduction and adaptation in any medium and for any purpose provided that it is properly attributed. For attribution, the original author(s), title, publication source (PeerJ) and either DOI or URL of the article must be cited.
License URL: https://creativecommons.org/licenses/by/4.0/

Keywords: Camera-trapping, Flood pulse, Floodplain dynamics, Wetlands, Várzea, Seasonal movements

Funding: The Brazilian Science Council (CNPq) Darwin Initiative for the Survival of Species grant DEFRA no. 20-001 Rufford Foundation Small Grants The Smithsonian Manson School of Conservation The School of Environmental Sciences at the University of East Anglia The Brazilian Science Council (CNPq) funded Hugo C.M. Costa’s MSc studentship at Museu Emílio Goeldi /UFPA. Fieldwork and equipment funding were provided by a Darwin Initiative for the Survival of Species grant (http://www.darwininitiative.org.uk/ DEFRA no. 20-001) awarded to Carlos A. Peres; the Explorers Club (https://explorers.org/); Idea Wild (http://www.ideawild.org/) and the Rufford Foundation Small Grants (http://www.rufford.org/12231-1). The Smithsonian Manson School of Conservation provided a partial scholarship award (http://smconservation.gmu.edu/ MCCS 0501) to Hugo C.M. Costa. The School of Environmental Sciences at the University of East Anglia (https://www.uea.ac.uk/environmental-sciences) funded Mark I Abrahams’ PhD studentship. There was no additional external funding received for this study. The funders had no role in study design, data collection and analysis, decision to publish, or preparation of the manuscript.

==============================
The flood pulse is the main factor structuring and differentiating the ecological communities of Amazonian unflooded (terra firme) and seasonally-flooded (várzea) forests as they require unique adaptations to survive the prolonged annual floods. Therefore, várzea and terra firme forests hammer out a spatio-temporal mosaic of resource availability, which may result in landscape scale seasonal movements of terrestrial vertebrates between adjacent forest types. Yet the lateral movements of terrestrial vertebrates between hydrologically distinct neighbouring forest types exhibiting staggered resource availability remains poorly understood, despite the important implications of this spatial dynamic for the ecology and conservation of forest wildlife. We examined the hypothesis of terrestrial fauna seasonal movements between two adjacent forest types at two contiguous sustainable-use forest reserves in Western Brazilian Amazonia. We used camera trapping data on the overall species richness, composition, and abundance of nine major vertebrate trophic guilds to infer on terrestrial vertebrate movements as a function of seasonal changes in floodplain water level. Species richness differed in neighboring terra firme forests between the high-and low-water phases of the flood pulse and terra firme forests were more species rich than várzea forests. There were clear differences in species composition between both forest types and seasons. Generalized Linear Models showed that water level was the main factor explaining aggregate abundance of all species and three trophic guilds. Our results indicate that the persistence of viable populations of large terrestrial vertebrates adjacent to major Amazonian rivers requires large, well-connected forest landscapes encompassing different forest types to ensure large-scale lateral movements by forest wildlife.

Introduction

Wetland habitats are both challenging to conserve and globally important for biodiversity conservation and human wellbeing (Keddy et al., 2009). Seasonal and perennial wetlands are exceptionally productive habitats that support both high densities and a high diversity of wild species (Halls, 1997; Junk et al., 2006). They also directly underpin the livelihoods of millions of people and provide ecosystem services including productive fisheries, water purification, hydrological regulation, nutrient cycling and naturally-fertilized agricultural land (Costanza et al., 1997; François et al., 2005). The associated seasonal movements of wetland fauna are especially challenging to conserve because their spatially complex life histories require resources provided by several distinct habitats and entail diverse anthropogenic threats at multiple sites (Martin et al., 2007; Wilcove & Wikelski, 2008).

A vast proportion of the Amazon Basin is formed by natural landscape mosaics of wetlands embedded within a matrix of upland (hereafter, terra firme) forests on generally nutrient-poor soils well above the maximum water-level of adjacent floodplains (Tuomisto et al., 1995). Amazonian floodplains comprise a variety of habitats including swamp forests, hydromorphic savannas, coastal wetlands, tidal forests, and seasonally-flooded forests. These Amazonian wetlands are classified according to their climatic, edaphic and floristic characteristics (Junk & Piedade, 2010; Junk et al., 2011). Based on these criteria, two large groups of wetlands have been broadly distinguished: those with either (i) relatively stable or (ii) oscillating water levels (Junk et al., 2011).

Most Amazonian wetlands with oscillating water levels are subjected to a predictable, long-lasting monomodal flood pulse which alternates between the high- and low-water periods according to the Flood Pulse Concept (Prance, 1979; Junk, Bayley & Sparks, 1989). Depending on the geomorphology and geochemical profile of each watershed, these areas can be inundated by white-, black- or clear-water rivers (Sioli, 1984). White-water rivers such as the Solimões, Madeira, Japurá and Juruá have their origins in the Andes or Andean piedmonts, are nutrient-rich, and have neutral pH. These rivers deposit their alluvial sediments along wide swaths of floodplain forests of high primary productivity, which are locally known as várzeas (Wittmann et al., 2006; Junk et al., 2011). In contrast, Amazonian black-water rivers such as the Negro, Tefé and Jutaí rivers discharge transparent-blackish waters with low suspended sediment loads and acidic pH. Forests inundated by black-water rivers are locally known as igapós and are typically supported by low-fertility soils and their trees exhibit 50% lower diameter increment compared to várzea forests (Junk & Piedade, 2010; Junk et al., 2011).

The flood pulse is the main factor structuring and differentiating the ecological communities of várzea and igapó forests from adjacent terra firme forests (Peres, 1997; Haugaasen & Peres, 2005a; Haugaasen & Peres, 2005b; Haugaasen & Peres, 2005c; Beja et al., 2009) as they require unique adaptations to survive the prolonged annual floodwaters. Terra firme forests are more species-rich, including more forest habitat specialists than várzeas and igapó, while the average population biomass density is higher in seasonally-flooded forests along white-water rivers (Peres, 1997). This predictable long-lasting and monomodal flood pulse triggers and synchronizes critical ecological events including the availability of plant reproductive parts (Nebel, Dragsted & Vega, 2001; Schöngart et al., 2002; Haugaasen & Peres, 2005a; Hawes & Peres, 2016), dietary shifts in primates, ungulates and fishes (Bodmer, 1990; Peres, 1994: Peres, 1999; Saint-paul et al., 2000), human extractive activities of non-timber forest products, and the exploitation of both terrestrial and aquatic prey (Newton, Endo & Peres, 2011; Endo, Peres & Haugaasen, 2016). As they are structurally and compositionally different, Amazonian várzeas, igapós and terra firme forests engender a spatio-temporal mosaic of resource availability which may result in landscape-scale seasonal movements of terrestrial vertebrates between these often neighbouring forest types (Bodmer, 1990; Peres, 1999; Haugaasen & Peres, 2007). Terra firme, várzea and igapó forests exhibit complementary fruit production peaks, whereby the fruiting peak in terra firme forests occurs during the onset of the wet season, whereas fruit maturation in várzeas and igapós begin during the late high-water season (Schöngart et al., 2002; Haugaasen & Peres, 2005a; Haugaasen & Peres, 2007; Hawes & Peres, 2016).

This asynchrony in fruit production attracts frugivorous fish and arboreal frugivores to floodplain forests during the high-water period (Saint-paul et al., 2000; Beja et al., 2009), whereas ungulates, carnivores, terrestrial insectivores and ant-following birds are attracted to várzeas and igapós immediately after the water level recedes. These lateral movements are due to the high abundance of fruit and seed deposited on the forest floor and higher insect abundance during this period (Bodmer, 1990; Peres, 1994; Adis & Junk, 2002; Haugaasen & Peres, 2007; Mendes Pontes & Chivers, 2007; Beja et al., 2009).

We tested the hypothesis that many terrestrial vertebrates move seasonally between Amazonian seasonally-flooded and unflooded forests by conducting camera-trapping surveys in both terra firme and várzea forests along a major white-water tributary of the Amazon river during both the high- and low-water phases of the flood pulse. We examined differences in vertebrate abundance, species richness, and changes in species composition between these two forest types and seasons. The contrast between the high- and low-water phases of the flood pulse was used to indirectly infer that the terrestrial fauna most likely leave terra firme forest and move into várzea forests during the low-water phase to take advantage of higher resource availability. Conversely, there should be transient overcrowding of the terrestrial vertebrate fauna in adjacent terra firme forests driven by lateral movements away from the rising floodwaters during the high-water phase. We provide crucial empirical evidence supporting the notion that Amazonian terra firme and várzea forests should be juxtaposed within fully functional floodplain protected areas, thereby enhancing both the spatial configuration of reserve design and landscape management of highly heterogeneous forest macromosaics in Amazonia for both biodiversity persistence and the subsistence of local extractive communities.

Materials & Methods

Study Area

This study was carried out at two contiguous sustainable-use forest reserves within the State of Amazonas, Brazil: the Médio Juruá Extractive Reserve (RESEX) spanning 253,227 ha, and the Uacari Sustainable Development Reserve (RDS) spanning 632,949 ha. Both reserves border the white-water Juruá River, the second largest white-water tributary of the Amazonas/Solimões River. These protected areas contain large expanses of terra firme forests (80% of both reserves) as well as an approximately 18.40 ± 5.71 km wide band of seasonally-flooded várzea forest (17.9%) encompassing the main river channel (Hawes et al., 2012) (Fig. 1). The Juruá region experiences an Af climate type (constantly humid) according to Köeppen criteria, with a mean annual temperature of 27.1 °C, a mean rainfall of 3,679 mm/year, and peak water levels of 14 m during a prolonged flood pulse, which is alternated by a dry phase in várzea between July and early November (Peres, 1997). All forest sites surveyed consist of largely undisturbed primary forest, although commercially valuable timber species have experienced non-mechanized selective logging along the Juruá River from 1970 to 1995, especially in várzea forests, which was banned since the formal creation of these two reserves.

Figure 1 Map of the study area in the central Rio Juruá region of western Brazilian Amazonia, Amazonas, Brazil.

Map inset shows the geographic location of the Juruá River and the study region. The boundaries of the RESEX Médio Juruá and RDS Uacari are outlined in black. Background colors represent elevation, with reddish and green shades indicating low and high elevation, respectively. Solid red circles represent camera trap stations (CTS) deployed radiating inland into terra firme forest (sample design 1). Green and aqua circles represent CTS deployed at terra firme forest sites near forest habitat boundaries along the várzea interface and far into várzea forest, respectively (sample design 2).

The RESEX Médio Juruá and RDS Uacari were created in 1997 and 2005, respectively, and are currently inhabited by some 4,000 legal residents, distributed across 74 local communities. These communities are located on both sides of the Juruá River, adjacent to either the main river channel or tributaries and oxbow lakes (Fig. S1). Residents of these reserves are variously engaged in agricultural and extractive activities for both subsistence and cash income (Newton, Endo & Peres, 2011; Campos-Silva & Peres, 2016).

Research permissions and full approval for this purely observational research were provided by Centro Estadual de Unidades de Conservação do Amazonas (CEUC/SDS/AM –020/2013) and by Instituto Chico Mendes de Conservação da Biodiversidade (ICMBio –38357-1).

Camera trapping

Data on the relative abundance of terrestrial vertebrates were collected at 279 camera-trapping stations (CTS) deployed at distances of 3,100 ± 367 m (x ¯±SD) apart, along a ∼514-km nonlinear section of the Juruá River (Fig. 1). We used Bushnell Trophy Cam 119436c, Reconyx Hyperfire HC500 and Bushnell 8MP Trophy Cam HD camera traps. These were programmed to record three and five consecutive photographs and 10-sec videos, respectively, at each trigger event without intervals. A CTS consisted of one camera trap deployed 40–60 cm above ground, and operated over a functional period of 38.7 ± 13.9 days (≈928.8 ± 333.6 h). The sensor sensitivity was set to high, and all CTS were unbaited and deployed away from trails.

Camera-trapping stations were deployed in two complementary sample designs (Table 1; Fig. 1): From April 2013 to June 2014, 193 CTS were deployed at intervals of 50 m, 350 m, 1,000 m, 3,000 m and 6,000 m Euclidean distance along transects, arrayed in contiguous terra firme primary forest, radiating away from local communities. This design facilitated surveys of terrestrial vertebrate abundance at varying distances from the várzea interface and at varying intervals during the receding flood pulse. In the second design, repeated over two inundation (March–April 2013 and 2015) and two low-water phases (September–October 2013 and 2014), CTS were deployed in both várzea forests and adjacent terra firme sites. In this arrangement, 30 terra firme CTS were deployed during both high- and low-water phases whereas 26 várzea CTS were surveyed only during the low-water phase, as várzea habitat is only available to the terrestrial fauna during this time of year. All várzea CTS were placed in high-várzea forests to avoid differences in plant species composition and phenology within sample sites (Wittmann et al., 2006; Parolin, Wittmann & Schöngart, 2010).

Table 1 Camera trapping effort at Amazonian flooded and unflooded forests, along the Juruá River, Amazonas, Brazil (see Fig. 1).

Sample Design	Flood pulse phase	Number of active CTS	
		Terra Firme	Várzea	
Sample design 1	From high to low water	193		
Sample design 2	High-water	30	–	
Sample design 2	Low-water	30	26	
Total		253	26	

Data management and estimates of the number of independent detections were undertaken using camtrapR version 0.99.8 (Niedballa et al., 2016). Images of conspecifics >30 min apart were defined as independent detection events. Species nomenclature followed the IUCN Red List (IUCN, 2018). Primates, non-terrestrial birds and rodents and marsupials smaller than 1 kg were excluded from our analyses, but all other avian and mammalian taxa were considered. Congener brocket deer (Mazama spp.), armadillos (Dasypus spp.), and small tinamous (Crypturellus spp.) were each treated as single species functional group due to difficulties in differentiating them in nocturnal (black and white) images.

Table 2 Covariates used to investigate the seasonal dynamics of terrestrial vertebrates in Amazonian flooded and unflooded forests, along the Juruá River region, western Brazilian Amazonia.

Covariate	Abbreviation	Description	
Area of várzea forest	vz0.5k	Area (m2) of seasonally flooded forest within a 500 m circular buffer centered at each CTS	
	vz1k	Area (m2) of seasonally flooded forest within a 1,000 m circular buffer centered at each CTS	
	vz5k	Area (m2) of seasonally flooded forest within a 5,000 m circular buffer centered at each CTS	
Distance to várzea forest	vzdist	Euclidean distance from each CTS to the nearest várzea forest	
Deforestation area	defor0.5k	Total area (m2) of deforestation within a 500 m circular buffer centered at each CTS	
	defor1k	Total area (m2) of deforestation within a 1,000 m circular buffer centered at each CTS	
	defor5k	Total area (m2) of deforestation within a 5,000 m circular buffer centered at each CTS	
Distance to nearest deforestation	defordist	Euclidean distance from each CTS to the nearest deforestation patch	
Community size	popcomm1	Number of residents of the local community nearest each CTS	
Distance to local community	commdist1	Euclidean distance from each CTS to the nearest local community	
Distance to urban center	citydist	Euclidean distance from each CTS to the nearest urban center	
Elevation	elev	Elevation (m) of the CTS above the main channel of the Juruá river.	
River distance	riverdist	Distance from each CTS to the midpoint of Juruá river	
Water level	waterlevel	Mean daily water level of the Juruá river during the deployment period of each CTS	

All species considered here were grouped into nine trophic guilds (frugivore-insectivores, granivore-frugivores, frugivores, carnivores, frugivore-carnivores, insectivore-frugivores, insectivores, browsers and frugivore-browsers) based on Benchimol & Peres (2015). An assemblage-wide metric of aggregate biomass was calculated by multiplying the species-specific camera-trap detection rate (number of detections/100 trap-nights) by the mean adult body mass per species, which could then be summed across all species detected at each CTS. For group-living species, we multiplied individual body mass values by the mean observed group size obtained from line-transect surveys conducted in the same study landscape (Abrahams, Peres & Costa, 2017).

For each CTS, we extracted landscape and human disturbance covariates using ArcGIS (version 10.3) (Table 2). We calculated the mean water level of the Juruá River during the exposure period of each CTS using daily water-level readings, recorded over 38 years (from 1st January 1973 to 31st December 2010; N ≈ 14,600 daily measurements) at a nearby locality (Gavião Metereological Station in Carauari-AM) (Fig. S2). As a continuous variable, mean water-level during CTS sampling intervals was a far more powerful descriptor of seasonality period than either categorical season (e.g., low-water vs high-water season) or time of the year (e.g., Julian day) per se.

Data analysis

All analyses were conducted in R version 3.3.2 (R Development Core Team, 2016). We first used both Student’s paired t-tests and ordinary t-tests to examine differences in species richness and abundance of terra firme forests between the high- and low-water phases, and between terra firme sites during the low-water phase and várzea forests, respectively. We estimated species richness per CTS, accounting for any differences in the number of trap nights, using a rarefaction method and first-order Jackknife estimator available in the specaccum function of the “vegan” package of R (Oksanen et al., 2013). We choose this estimator because it gives the most reliable results in tropical forest camera-trap studies (Tobler et al., 2008). For the abundance analyses, we considered the camera-trapping rate (number of independent detections per 100 trap-nights) as our response variable. These analyses were performed using CTS data from our second sample design, which targeted from both terra firme forests during the high- and low-water phases of the flood pulse and várzea forests during the low-water period.

Principal Coordinates Analysis (PCoA) was used to visually depict variation in vertebrate assemblage structure. Differences in assemblage structure between both forest types and seasons were tested using Permutational Multivariate Anova (PERMANOVA) (Anderson, 2001) with two factors with two levels each. Prior to these analyses, to reduce the weight of excessively abundant species in the ordination space, terrestrial vertebrate abundance was standardized by dividing the number of detections of each species by the total number of detections at each CTS. PCoA and PERMANOVA were performed using a Bray–Curtis similarity distance matrix derived from both of our sample designs. To test for seasonal effects on species composition at terra firme CTS, we performed a Procrustes rotation analysis of the Bray-curtis ordination matrices derived from CTS from our second sample design addressing both the high- and low-water phases of the flood pulse.

We tested the hypothesis of seasonal faunal movements between adjacent forest types and seasons by investigating the effects of river water level on the overall species abundance, species richness, overall vertebrate biomass, and on the number of captures of the nine trophic guilds. We controlled for the effects of landscape context and anthropogenic disturbance that may deplete wildlife populations near human settlements across the study area (Abrahams, Peres & Costa, 2017) by including these variables in the analysis. We employed Generalized Linear Models (GLMs) using a Poisson distribution for count data using the combined CTS from both sample designs, but a Negative Binomial distribution was chosen when overdispersion was detected (Hilbe, 2007). For our metric of biomass, we used a Gaussian error structure. The number of camera-trapping nights per CTS was specified as an offset variable in all models to account for difference in sampling effort (i.e., number of active days/nights) between CT deployments.

We controlled for high levels of variable inter-dependence by performing a Pearson’s correlation matrix, retaining non-correlated variables (r < 0.70). We retained 11 variables describing the local habitat, season, landscape context, and level of human disturbance of CTS sites (vz1k, vzdist, elev, waterlevel, riverdist, defor1k, defor5k, defordist, ctydist, popcomm1 and commdist1; see description of these variables in Table 2). For those variables representing the same class of human disturbance (e.g., deforestation area), the appropriate buffer size was determined by running all models using different buffer thresholds, and then using the threshold resulting in the strongest effect on our response variables. We mitigated for collinearity between the predictors using the Variance Inflation Factor (VIF < 3), excluding the variables above this threshold. We used Akaike’s Information Criteria (AICc) to select the models that best fit the data, employing a stepwise method starting with the full model and discarding predictors until we reached a model with the lowest AICc value. In these models we used data from both of our sample designs.

Results

On the basis of 10,447 trap-nights, we recorded 4,059 independent detections of 25 terrestrial vertebrate species, including 21 mammals representing 12 families and eight orders and four large-bodied bird species (Table 3). We found clear differences in terra firme forest sites in both species richness and abundance between high- and low-water phases (richness: paired t = 2.552, df = 21, p = 0.018; abundance: paired t = 2.950, df = 21, p = 0.007, Figs. 2A, 2C). During the low-water season, overall abundance was higher in terra firme than in várzea sites (t = 2.709, df = 48, p = 0.009, Fig. 2B). Similarly, species richness was higher in terra firme sites (18.42 ± 3.11 species) than in adjacent várzea sites (14.31 ± 3.00 species; t = 4.748, df = 48, p < 0.001, Fig. 2D).

Table 3 Terrestrial vertebrate species detected by camera trapping stations (CTS) deployed in this study in Amazonian flooded and unflooded forests, along Juru river, Amazonas, Brazil.

Class	Order	Species	English vernacular name	Trophic guild	
AVES	GRUIFORMES	Psophia leucoptera (Spix, 1825)	Pale-winged trumpeter	Frugivore-Insectivore	
	STRUTHIONIFORMES	Crypturellus spp (Brabourne & Chubb, 1914)	Small tinamous	Granivore-frugivore	
	GALLIFORMES	Tinamus sp (Hermann, 1783)	Great tinamous	Granivore-frugivore	
		Mitu tuberosum (Spix, 1825)	Razor billed curassow	Frugivore	
MAMMALIA	CARNIVORA	Panthera onca (Linnaeus, 1758)	Jaguar	Carnivore	
		Procyon cancrivorus
(G.[Baron] Cuvier, 1798)	Crab-eating-racoon	Frugivore-insectivore	
		Puma concolor (Linnaeus, 1771)	Puma	Carnivore	
		Herpailurus yagouaroundi
(É. Geoffroy Saint-Hilaire, 1803)	Jaguarundi	Carnivore	
		Leopardus wiedii (Schinz, 1821)	Margay	Carnivore	
		Leopardus pardalis
(Linnaeus, 1758)	Ocelot	Carnivore	
		Speothos venaticus (Lund, 1842)	Bush dog	Carnivore	
		Eira barbara (Linnaeus, 1758)	Tayra	Frugivore-Carnivore	
		Atelocynus microtis (Sclater, 1883)	Small-eared-dog	Frugivore-Carnivore	
		Nasua nasua (Linnaeus, 1766)	Coati	Frugivore-insectivore	
	CINGULATA	Priodontes maximus (Kerr, 1792)	Giant armadillo	Insectivore-Frugivore	
		Dasypus spp (Linnaeus, 1758)	Armadillo	Insectivore-Frugivore	
	CETARTIODACTYLA	Tayassu pecari (Link, 1795)	White lipped peccary	Granivore-Frugivore	
		Pecari tajacu (Linnaeus, 1758)	Collared peccary	Granivore-Frugivore	
		Mazama spp (Rafinesque, 1817)	Brocked deer	Browser	
MAMMALIA	PERISSODACTYLA	Tapirus terrestris (Linnaeus, 1758)	Tapir	Browser	
	PILOSA	Tamandua tetradactyla (Linnaeus, 1758)	Southern tamandua	Insectivore	
		Myrmecophaga tridactyla (Linnaeus, 1758)	Giant anteater	Insectivore	
	RODENTIA	Myoprocta pratti
(Pocock, 1913)	Green acouchy	Granivore-frugivore	
		Dasyprocta fuliginosa Wagler, 1832	Black agouti	Granivore-frugivore	
		Cuniculus paca
(Linnaeus, 1766)	Paca	Frugivore-browser	

Figure 2 Comparison between terra fime and várzea forests during both the high- and low-water phases of the flood pulse considering both the total abundance and species richness of terrestrial forest vertebrates.

Boxplots comparing abundance and rarefied species richness between terra firme forests during both high- (dark green) and low-water (light green) phases of the flood pulse (A and C) and between várzea (orange) and terra firme forests (light green) during the low-water phase (B and D).

At terra firme sites, the black agouti (D. fuliginosa) was the most common species followed by the brocket deer (Mazama spp), pale-winged trumpeter (P. leucoptera), razor-billed curassows (M. tuberosum) and collared peccaries (P. tajacu). The detection rates of these species were higher during the high-water season than during the low-water season, whereas pacas (C. paca), jaguars (P.onca), giant anteaters (M. tridactyla), giant armadillos (P. maximus) and tapirs (T. terrestris) were more frequently detected during the high-water phase (Fig. 3A). During the low-water season, brocket deer, black agoutis, pacas, pale-winged trumpeter, razor-billed curassows and collared peccaries were more abundant in terra firme than in adjacent várzea forests, while tapirs, ocelots (L. pardalis), pumas (Puma concolor) and small tinamous (Crypturellus spp) presented higher detection rates in várzea (Fig. 3B).

Figure 3 Camera trapping rate of terrestrial vertebrates recorded in terra firme and várzea forests.

(A) Camera trapping rates in terra firme forests during both high- (dark green bars) and low-water phase of the flood pulse (light green bars). (B) Camera trapping rates in both terra fime and in várzea forests during the low-water phase of the flood pulse. Light green and orange bars represent terra firme and várzea forests, respectively. Species are represented by the first four letters of each genus and first four letters of each species and ordered from least to most abundant top to bottom. Asterisks indicate significant differences according to paired (A) and unpaired t-tests (B); *p ⩽ 0.05, ** p ⩽ 0.01, ***p ⩽ 0.001.

PCoA ordination revealed differences between sample clusters formed by all terra firme sites between the high- and low-water phases of the flood pulse, and between várzea forests and terra firme sites during the low-water phase (Fig. 4A), which was further confirmed by permutation tests (PERMANOVA; F = 3.964, p = 0.002; F = 10.401, p = 0.001, respectively). Terra firme sites occupied the largest area in community space during the high-water phase, with both terra firme and várzea forest sites during the low-water phase occupying subsets of the larger group, and várzea sites occupying the smallest area. Additionally, the Procrustes rotation performed with the terra firme CTS from sample design two indicated significant differences in ordination space in the multivariate structure of community composition between the high- and low-water phases (R = 0.74, p = 0.007, Fig. 4B).

Figure 4 Terrestrial vertebrate species composition in Amazonian seasonally-flooded and unflooded forests during both high- and low-water phases of the flood pulse.

(A) Principal Coordinates Analysis (PCoA) ordination of the terrestrial vertebrate assemblage structure detected by camera traps in Amazonian terra firme forests during both high- and low-water phases of the flood pulse (green and light-green circles, respectively) and in várzea forests (orange circles). (B) Procrustes rotation plot of terra firme sites sampled during both high- and low-water phase of the flood pulse. Arrows (vectors) indicate the species migration in community space from the high- to the low-water season.

Generalized linear models (GLMs) revealed that water level was a significant positive predictor of both overall species abundance and the detection rates for three trophic guilds: frugivore-insectivores, granivore-frugivores and carnivores (Figs. 5A, 5D, 5F, 5G). The size of the nearest local extractive community was associated with higher detection rates for browsers (Fig. 5J). Likewise, elevation was a positive predictor of detection rates of insectivore-frugivores (Fig. 5I). The best model for frugivores retained only elevation as a significant negative predictor (Fig. 5E). The area of várzea within a 1,000-m buffer around each CTS best explained insectivore detection rates (Fig. 5L), while distance to the nearest urban center had the opposite effect on our metric of overall vertebrate biomass (Fig. 5B). The best GLM model explaining overall species richness and the detection rates of frugivore-carnivore and frugivore-browsers failed to retain any significant predictors (Figs. 5C, 5H, 5K).

Figure 5 Coefficient estimates (± 95% confidence intervals) showing the magnitude and direction of effects of different explanatory variables retained in the best performing GLMs.

(A) aggregate abundance, (B) aggregate biomass of all species, (C) species richness (D–L) numbers of detections of each trophic guild.

Discussion

Species richness, composition and seasonal movements between forest types

Our camera-trapping study provides tantalizing evidence that water level governs the distribution of large terrestrial vertebrates in Amazonian pristine forest mosaics. These species appear to exhibit lateral seasonal movements to take advantage of periodic resource availability in extremely productive floodplain forests. In our study area, the swath of floodplain forest is approximately 20-km wide, thereby providing a vast area of highly productive habitat for terrestrial species during the low-water phase.

In general, terra firme forest sites were more species-rich than várzea forest sites, a pattern that conforms with results from previous studies comparing assemblages of all mammals, primates, bats, birds and small mammals in Amazonian seasonally-flooded and unflooded forests (Peres, 1999; Peres, 1997; Haugaasen & Peres, 2005b; Haugaasen & Peres, 2005c; Beja et al., 2009; Pereira et al., 2009; Bobrowiec et al., 2014). Salvador, Clavero & Leite Pitman (2011) reported that floodplain forests in the Peruvian Amazon are more species-rich than terra firme forests during the dry season, which is contrary to our findings. This can be explained by methodological differences between the studies once they used line transects, track counts and interviews enabling the inclusion of semi-aquatic and arboreal mammals such as giant otters, primates and sloths in their dataset. They also report that the number of species in floodplain forest during the wet season remains the same throughout the year, while in terra firme, a sharp increase in species richness coincided with the onset of the wet season. These shifts in species richness between the two forest types are consistent with our seasonal movement hypothesis, as many terrestrial vertebrate species likely exit terra firme terrains to take advantage of seasonally abundant food resources in várzea forest.

Water level represents a physical barrier for most vertebrate species attempting to access várzea forests during the high-water phase. This was confirmed by the positive relationship between water level and aggregate community-wide abundance, and the number of detection events of frugivore-insectivores, granivore-frugivores and carnivores. Bobrowiec et al. (2014) noted that the flood pulse constituted a physical barrier even for Phyllostomid bats, whose species composition differed between terra firme and várzea forests during the high-water period, but this effect did not persist year-round. We found clear differences in species composition between terra firme and várzea forests during the low-water phase and within our terra firme samples between the high- and low-water phases of the annual cycle. These results imply that forest fauna can exhibit ephemeral occupancy of várzea sites during the dry season and that the rising flood waters force several species to seek suitable habitats in upland forests. These seasonal lateral movements drive differences in species richness and composition between both seasons and forest types.

Food availability and its distribution within forest habitats, is the most important variable explaining the occupancy and abundance of mammals in different forest types (Mendes Pontes, 2004; Haugaasen & Peres, 2007). In terra firme forests, fruit production occurs during the early wet season whereas in várzea forests, fruit production starts during the late wet season (Hawes & Peres, 2016). A substantial proportion of the large terrestrial fauna may therefore move between várzea and terra firme forests to exploit seasonally available resources. For instance, frugivore species in our models exhibited a negative abundance relationship with terrain elevation. This predictor can be used to distinguish both forest types, as our terra firme CTS were on average situated on terrains 14 m higher than our várzea CTS (t-value = 9.458, df = 277, p-value < 0.001). As water levels recede, the terrestrial fauna rapidly colonize várzea forests to forage on the seasonal production of residual fruit- and seed-fall (total production minus dispersal and consumption by arboreal frugivores), which can be twice as high as in adjacent terra firme forests during this period (Bodmer, 1990). Ungulate species such as collared peccaries and brocket deer exhibit a marked dietary shift following the flood pulse, consuming more fruits in seasonally-flooded forests during the low-water period compared to the high water period (Bodmer, 1990).

Water level is an important determinant of species detection rates in highly heterogeneous forest landscapes subjected to marked seasonal floods (Negrões et al., 2011; De Lázari et al., 2013). Haugaasen & Peres (2007) reported three different strategies of landscape movements across forest types, which were reflected in our results: wide-ranging species, year-round residents and interface species. Large-bodied granivore-frugivores such as the large-group-living white-lipped peccaries is a wide-ranging “landscape” species that, on a seasonal basis, occupies large home ranges in different forest types and shift their diets and habitat use in response to both seasonal flooding and resulting resource fluctuations (Bodmer, 1990; Fragoso, 1998; Keuroghlian, Eaton & Desbiez, 2009). Large-bodied myrmecophages and insectivore-frugivores such as giant anteaters and armadillos exhibited low detection rates in várzea forests, likely because they are year-round residents in terra firme forests, which was confirmed by the negative relationship in our models between terrain elevation and the detection rates of these species. They are also less likely to move between forest types because the permanently wet várzea soils preclude their fossorial foraging behavior. We never observed giant armadillo (P. maximus) holes in várzea forests, but commonly observed them in terra firme forests, and this is consistent with previous studies in the Araguaia River (Negrões et al., 2011) and Peruvian floodplain forests (Salvador, Clavero & Leite Pitman, 2011).

Detection rates of carnivores increased with the water level, a pattern that can be explained by their swimming and climbing abilities, which allow them to both move between temporary forest islands and utilize the tree canopy as floodwaters rose. Jaguars (P. onca) in várzea forests in the lower Japurá River are known to spend the entire high-water season high up in the trees (EE Ramalho, pers. comm., 2014) and subsist upon arboreal and semi-aquatic species such as howler monkeys (Alouatta seniculus (Linnaeus, 1766)), sloths (Bradypus variegatus, Schinz, 1825) and spectacled and black caimans (Caiman crocodilus (Linnaeus, 1758), and Melanosuchus niger (Spix, 1825)) (Ramalho, 2006).

Conservation implications

Our research supports the existing body of evidence that the Médio Juruá region, and many other regions of the lowland neotropics, should be viewed as an essentially interconnected multi-habitat socio-ecological system. The massive long-lasting seasonal flood pulse (Junk, Bayley & Sparks, 1989) and the associated phenological (Hawes & Peres, 2016), hydrological, ecological (Hawes et al., 2012) and livelihood impacts this engenders (Endo, Peres & Haugaasen, 2016) require conservation planning at the scale of the entire landscape, with major drainage basins representing complementary management units.

Várzea and terra firme forests function as ecologically integrated and hydrologically interconnected habitats that are seasonally utilized by a suite of mobile species, with terrestrial fauna often relying upon the temporally staggered resources of both habitats. As such, they are threatened by both aquatic and terrestrial anthropogenic activities at the local and regional scales. The immense fluvial transport network of the lowland Amazon makes even remote forests accessible to hunters (Peres & Lake, 2003), making their faunal resources non-excludable, whilst simultaneously difficult to monitor.

The existing protected area network and management policies in Amazonian seasonally-flooded forests were created principally to protect terrestrial ecosystems and therefore suffer from design, implementation and monitoring deficiencies and their delimitations does not adequately represent or protect the full suite of biotic diversity (Peres & Terborgh, 1995; Albernaz et al., 2012; Castello et al., 2013). Although a protected area coverage of ∼25% gives the impression of extensive conservation management of floodplains, less than 1% of the aggregate area of Amazonian floodplains in Brazil is strictly protected (Albernaz et al., 2012). Sustainable development and extractive reserves represent the majority of all floodplain protected areas. Their conservation effectiveness can be compromised by high human population density, the uncertain economic viability of exploiting non-timber resources and a shortfall in available animal protein resulting from depleted game vertebrate populations (Peres, 2011; Terborgh & Peres, 2017), but see Abrahams, Peres & Costa (2017) and Campos-Silva & Peres (2016) for best-case scenarios of terrestrial subsistence hunting and local fisheries management.

We have shown that a substantial part of the large vertebrate fauna modulates their use of different forests types within a highly heterogeneous forest landscape according to the marked seasonality of várzea floodplain forests. Our study represents the confluence between the issues of landscape-scale conservation planning, ecological connectivity, nutrient transport and uptake, and community-based natural resource management. The Médio Juruá region exemplifies these issues as it encompasses extensive seasonal wetlands and a suite of hunted, seasonally-mobile species. Adequate conservation strategies in this region must account for the full life-history needs of mobile harvested species, ecologically interconnected habitats and the diverse livelihood portfolios of local communities (Lindenmayer et al., 2008). Different Amazonian forest types exhibiting staggered resource pulses must be included within the same or neighboring sustainable-use protected areas. This will provide sufficiently large areas to both support large-scale ecological processes (e.g., species migrations, lateral movements, persistence of apex predators) and anthropogenic extractive activities in the long run (e.g., estimated sustainable harvest area for tapir populations >2,000 km2) (Peres & Terborgh, 1995; Peres, 2001; Peres, 2005; Haugaasen & Peres, 2007). This concept can be applicable to conservation planning of other regions consisting of natural forests mosaics experiencing seasonal floods such as the hyper-fragmented region of the Araguaia River or at the Pantanal floodplains (Negrões et al., 2011; De Lázari et al., 2013). In these different scenarios, private reserves must be situated adjacent to protected areas to ensure terrestrial fauna protection during the prolonged inundation season.

Study limitations

In our study, we were unable to estimate the species richness in várzea forests during the high-water phase of the flood pulse, because our camera trapping method focused only on terrestrial species, which are more sensitive to the flood pulse than arboreal and semi-aquatic species. Várzea forests along this section of the Juruá River are typically subjected to an annual flood pulse amplitude of 8 to 12 m, which lasts for up to six months. Any camera traps deployed in várzea forests during the high-water period would need to be placed almost half way up into the forest canopy.

We acknowledge that these landscape-scale seasonal movements between forest types can only be conclusively verified by either radio or GPS telemetry studies targeting multiple species. The prohibitive costs of such an undertaking limit its community-wide feasibility. Our evidence is based on patterns of local population abundance, species richness and biomass, particularly along the várzea - terra firme interface, where temporary overcrowding is expected to occur for species abandoning the wide belt of várzea forest during the rise of floodwaters.

Conclusions

The annual floodwaters along several major white-water rivers in the Amazon is the main factor structuring and differentiating várzea floodplains from adjacent terra firme forests as unique adaptations are required to tolerate the prolonged flood pulse. This remarkable natural phenomenon drives several key ecological processes, including staggered plant phenology, high plant productivity, and supports major local livelihood activities such as subsistence fishing and hunting. This landscape scale seasonal dynamics between these major adjacent forest types was investigated in terms of species richness, species composition and population abundance for as many as 25 vertebrate species. We have shown that many upland forest terrestrial vertebrate species make seasonal use of várzea forests to take advantage of the abundant trophic resource in this forest type following the receding waters. We acknowledge that detailed movement data using GPS telemetry can further clarify the magnitude and seasonal importance of várzea habitat use by terra firme vertebrates. However, we highlight that this unique seasonal dynamic is a critical issue in Amazonian forest reserve design and biodiversity monitoring, particularly within large sustainable use reserves encompassing complex natural landscape mosaics, where unimpeded lateral movements should continue to support both local extractive economies and healthy wildlife populations.

Supplemental Information

Supplemental Information 1 Camera trap raw data

Number of detections of 25 terrestrial vertebrate species recorded across the middle Juruá region in unflooded and flooded forests

Click here for additional data file.

Supplemental Information 2 Camera trap raw data metadada

Abbreviations and Linnean binomials of 25 terrestrial vertebrate species recorded across the middle Juruá region in unflooded and flooded forests

Click here for additional data file.

Supplemental Information 3 Camera trap variables

Landscape and human disturbance variables extracted from 279 camera trap stations in the middle Juruá region, Amazonas, Brazil.

Click here for additional data file.

Supplemental Information 4 Camera trap variables metadata

Abbreviations and description of landscape and human disturbance variables extracted from 279 camera trap stations in the middle Juruá region, Amazonas, Brazil.

Click here for additional data file.

Figure S1 Map of the study area in the central Rio Juruá region of western Brazilian Amazonia, Amazonas, Brazil

Map inset shows the geographic location of the Juruá river and study region. The boundaries of the RESEX Médio Juruá and RDS Uacari are outlined in black. Seasonally flooded forests and terra firme forests are represented in light and dark gray respectively. Solid red circles represent camera trap stations (CTS) deployed radiating inland into terra firme forest (sample design 1). Green and aqua circles represent CTS deployed at terra firme forest sites near forest habitat boundaries along the várzea interface and far into várzea forest, respectively (sample design 2). Blue pentagons represent the location of human settlements.

Click here for additional data file.

Figure S2 Mean water level of the Juruá River obtained from daily readings recorded over 38 years (from 1st January 1973 to 31st December 2010) at Gavião Metereological Station in Carauari-AM

Click here for additional data file.

We are deeply grateful to the local communities of the Juruá region for their hospitality and friendship during fieldwork and to Gilberto Olavo from the Centro Estadual de Unidades de Conservação do Amazonas (CEUC/SDS/AM); and Rosi Batista and Manoel Cunha from the Instituto Chico Mendes de Conservação da Biodiversidade (ICMBio) for permitting our research work. We thank A. C. Mendes-Oliveira, F. Michalski, F. Palomares, and N. Negrões-Soares for their comments on previous versions of the manuscript. This publication is part of the Projeto Médio Juruá series on Resource Management in Amazonian Reserves (http://www.projetomediojurua.org).

Additional Information and Declarations

Competing Interests

Author Contributions

Animal Ethics

Data Availability

The authors declare there are no competing interests.

Hugo C.M. Costa conceived and designed the experiments, performed the experiments, analyzed the data, contributed reagents/materials/analysis tools, prepared figures and/or tables, authored or reviewed drafts of the paper, approved the final draft.

Carlos A. Peres and Mark I. Abrahams conceived and designed the experiments, performed the experiments, contributed reagents/materials/analysis tools, authored or reviewed drafts of the paper, approved the final draft.

The following information was supplied relating to ethical approvals (i.e., approving body and any reference numbers):

Centro Estadual de Unidades de Conservação do Estado do Amazonas (CEUC/SDS/AM) and Instituto Chico Mendes de Conservação da Biodiversidade ICMBio, provided full approval for this purely observational research.

The following information was supplied regarding data availability:

The raw data are provided in the Supplemental Files.

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
