# Peer review of "Seasonal dynamics of terrestrial vertebrate abundance between Amazonian flooded and unflooded forests"

_PeerJ, doi:10.7717/peerj.5058_

## Round 0.1 · original submission · Minor Revisions

The three reviewers all found the research to be valuable and relevant, but raised some concerns with regard its presentation in the manuscript and recommended either minor (2 reviewers) or major (1 reviewer) corrections. Having read the manuscript, I consider that minor revisions are needed, since these relate to clarifications or modifications within the manuscript text, rather than reworking of data or analysis. The changes suggested will help to produce a tighter, more focused paper. Please address the points raised in the reviews themselves, making sure in particular to:

• Consider reorganising the structure and/or content of the paper in order to (i) make clearer the gap in knowledge and research questions or hypotheses addressed; (ii) describe the usefulness of the analysis for addressing these questions; and (iii) revise the discussion of the conservation implications to be clearer, more concise, and directly relevant to your research as presented.
• Carefully reword statements in the abstract and introduction which pertain to the hypotheses tested, acknowledging that the camera trap methods employed do not allow for detection of movements, per se, but rather you are inferring movements from the data.
• In the methods section, make the link between the research objectives and methods used clearer, and provide important details and clarifications regarding sampling design (including details of the selection of camera trap site locations) and the GLMs used.
• Provide more details of the statistical methods used (in particular PCoA and PERMANOVA), how they were implemented for your dataset and why; at the start of the data analysis section, it would be helpful if you provided a brief summary of what follows - i.e. the structure of the analysis which was performed.
• Consider whether it may be beneficial to focus the manuscript by excluding anthropogenic disturbance as an additional factor in your analyses, since it is difficult to assess and quantify, may not provide clear benefits to the research, and since it detracts from the main message of the paper; alternatively, more details should be provided in both the results and discussion, as well as justification for your method of defining anthropogenic disturbance; supporting information (e.g. maps of deforestation, settlement locations) would also be needed.
• Address concerns regarding the discussion, particularly with regard to clarity and the issues raised regarding contrary or contradictory results presented in other papers – your discussion should include possible reasons for differences found.
• Include consideration and discussion of the movement of vertebrates into the inventoried region from up- or downriver, etc.
In addition, please:
• Address language issues raised; consider the use of a good proof-reader.
• As suggested, consider using a repository to share images from the camera-trap study.
• Present key variables (e.g. water level) and findings (e.g. trap rate, species richness) as maps; the location of várzea and terra firme forest areas should also be indicated.

·

Basic reporting

While the authors repeatedly state on the investigation of the hypothesis of seasonal movements of terrestrial invertebrates between the investigated forest types, their research design does not entirely allow for a true test of this hypothesis. The camera trap stations monitor abundance and richness data allowing for spatial and temporal comparison, but do not allow for the detection of movements. While I acknowledge that the results can be interpreted in that way, I suggest rewording the related statements in the abstract and the Introduction (i.e., L10-22, L113-114), which could make the general scope of the manuscript much stronger. In the end, vertebrates can move within the single forest types, entering the inventoried region from up- or downriver, etc., and these movements are not considered nor discussed.

Experimental design

As the primary scope of the manuscript is crystal-clear (investigate spatial and temporal differences in abundance and richness, and relate them to abiotic variables such as elevation and flood pulse), I wonder why the authors tried to incorporate anthropogenic disturbance as an additional factor in in their analyses (L118-120). Anthropogenic disturbance is difficult to assess and quantify, and I cannot see the benefits of including it here. Rather, I think simply excluding it would give more emphasis to the main result and message of the paper. If the authors decide to leave it in, they should also describe the results on that in more detail and include some discussion on it.

In the second sample design (L164-169), it is not clear how many CTS were established in flooded and how many CTS were established in terra firme forest. It would also be good to know if CTS established in varzea were located at the same or different elevations (= flood levels), as I expect huge floristic differences (=resource differences) between highly flooded and less flooded forest types within varzea. For example, phenological peaks during the high water season is mostly concentrated on highly inundated trees (low-varzea trees), while fruiting of less inundated (high-varzea) trees is much less synchronized with the flood level and season (i.e. Parolin et al. 2010, book chapter ed. by Junk et al. 2010, Springer).

Validity of the findings

I found the section on conservation implications a bit long and wordy. In addition, not all information given here is directly related to your results. I think this topic could be significantly shortened.

Additional comments

Some minor comments:
L120-124: A good reference here would be from Sabo et al. (2005, Ecology 86:56-62): “Thus conservation planners can easily increase the number of species protected in a regional portfolio by simply including a river within terrestrial biodiversity reserves”.

L133-134: The statement about the climate classification can be deleted.

L135-137: Not sure about the numbers you present here “…peak water levels of 8-12 m”. You mean: flood amplitude is 8-12 m? Please state exact numbers from the water gauge.

L137: 65-170 m above sea level includes terra firme, right? Please state the elevational range for varzea and terra firme separately here.

L139: Not sure about “small-scale selective logging”. In fact, all varzea forest is (intensely) selectively logged even if far away from bigger cities. You may better state that logging is impeded since the establishment of the RDS.

L147-149: I would place this info in the acknowledgements, or in a separate section at the end of the manuscript.

L321: …and consumption by fish and other aquatic foragers.

L394ff: “…communities with greater access to terra firme are more agricultural….” Don’t agree. Just the crops change. Usually, agriculture is much more intense in varzea than it is in terra firme because of the nutrient-rich soils that allow for several crops per year. See Junk et al. (2000), or McGrath et al. (2005, PLEC News and Views).

L398: Heavea spp. should be Hevea spp.

Fig.1: Instead of low and high elevations, you should give the exact altitudinal range for the colors used in the map.

·

Basic reporting

The reporting is clear and professional. However, a quick check by a good proof reader could take out some typos, such as choose for chose, exploitating for exploiting, and at for on.

The derived data is shared, but in common with most camera-trap studies, the images are not. These could be deposited somewhere.

Experimental design

Within the logistical limits, the design is good.

Validity of the findings

The findings are straight forward and the conclusions well stated, with caveats when necessary.

Additional comments

The study involved a great deal of work and is well done. You might like to look at the paper by Lyn Branch (Primates 24(3):424-431), which indicated that some species move into the várea at high water.

Reviewer 3 ·

Basic reporting

- Clear and unambiguous, professional English used throughout: no comments.
- Literature references, sufficient field background/context provided: Yes
- Professional article structure, figs, tables. Raw data shared: Yes
- Self-contained with relevant results to hypotheses: Yes

Experimental design

- Original primary research within Aims and Scope of the journal: Yes
- Research question well defined, relevant & meaningful. It is stated how research fills an identified knowledge gap: No. This point presents some problems. See General Comments to the authors.
- Rigorous investigation performed to a high technical & ethical standard: Yes.
- Methods described with sufficient detail & information to replicate: Yes.

Validity of the findings

- Impact and novelty not assessed: This point should be improved (See comments)
- Data is robust, statistically sound, & controlled: Yes.
- Conclusion are well stated, linked to original research question & limited to supporting results: There are some problems with this point (See comments to the author).
- Speculation is welcome, but should be identified as such: Some problems with this point and for implications for conservation (See comments to the authors).

Additional comments

I have carefully read the manuscript #25773. The authors carried out an evaluation of the migration of the fauna in a forest system that is under the stress of seasonal flooding. I believe the general aim of the manuscript is in line with the objectives of the journal. Moreover, the study area deserves special attention, given its important biodiversity and accelerated environmental degradation.

However, I recommended a reorganization of the structure and content of the paper in order to facilitate its understanding. For example, authors must add clear mentions of (1) the problem or gap of knowledge (Introduction), (2) the hypothesis/questions (Introduction), (3) the different predictions (Introduction), (4) the usefulness of each analysis or sampling design to answer each question. Moreover, a straightforward discussion about the implication of the results to conservation is needed.

I addressed major comments in this text, and some minor comments in the PDF. (English is not my mother tongue; please excuse any errors on my part).

Abstract
1. Line 20. "We examined the hypothesis of seasonal movements between two adjacent forest..". Whose movements?
2. Lines 23-24. "Species richness differed in neighboring terra firme forests between the high-and low-water phases of the flood pulse". Please, specify what the difference was (higher richness during the high-water phase?).

Introduction
1. The second paragraph (about the hunting problem of migratory species) felt very disconnected from the previous one (which talks about the importance of wetlands) and from the next (which talks about the flooded systems in the Amazon).
2. The gap of knowledge, problem statement or justification of the importance of this study, is not well addressed in the Introduction. For example, the two paragraphs preceding the Objective are basically an explanation (with facts and citations) of the lateral movements of the fauna between varzea and terra firme. Curiously, the abstract did mention (although imprecisely) the existence of a gap on knowledge.
3. Lines 105-109. The authors mentioned the prediction, but not the hypothesis. I believe the hypothesis is hide in this paragraph, but it is important to make it clear.
4. Lines 115-117 "To test this hypothesis we conducted camera-trapping surveys in both terra firme and várzea..." Again, what hypothesis? The existence of lateral movement?
Methods
1. I think that a clearer structure and explanation of the link between the questions/predictions/hypothesis and the design/analysis is needed. In general, the Methods felt like a great amount of tests, whose purposes are difficult to detect.
For example, the Design 1 (160-162), which compares terra firme fauna during both periods and uses a paired t-test, makes sense to me. With this analysis we can test if the prediction was correct (i.e. high richness in terra firme during the high-water phase). However, it is not clear why did the authors use the distance intervals in the sampling. It was for evaluating the effect of anthropogenic threats (a secondary objective)? Or did they expect high richness in sites closer/farther from the varzea interface?
On the other hand, the Design 2 only compares terra firme and varzea during the low-water phase (Lines 164-166). Thus, I am not sure about how this design can test the prediction of "a higher richness in terra firme (only) during flooding than during low water level" or to test the hypothesis of lateral movement (if this is the hypothesis). What were the expected results from this design? What if the richness in terra firme were lower than in varzea during the low-water phase? I suspect that this design is more useful to understand how the species migrate among forests. In any case, it would be helpful to describe the purpose of this analysis in Methods or to write the specific question that this design will answer.
2. Lines 218-220. "We tested the hypothesis of seasonal faunal movement between forest types and investigated the effects of landscape context and anthropogenic disturbance employing Generalized Linear Models (GLMs)". What are the response variables? How did the authors test the hypothesis of the existence of lateral movement (if this is the hypothesis) by doing these GLMs? By testing if the water level is selected as predictor of species richness in terra firme? Please, clarify this.

Discussion
1. Line 218. Here is the first description of a hypothesis.
2. Lines 290-291: "Salvador, Clavero & Leite Pitman (2011) reported that floodplain forests in the Peruvian Amazon are more species-rich than terra firme forests during the dry season". This result is contrary to yours (lower richness in flooded forests). Why? Is this difference important?
3. Line 305-307. "These results imply that forest fauna can exhibit ephemeral occupancy of várzea sites during the dry season...". With this sentence the purpose of Design 2 is a little clearer. I don't know if I am right, but for me the Design 2 is more related to testing if species from terra firme forests are also present in varzea, which may support (although not confirm) the existence of species migration. In other words, I believe the importance of Design 2 is more about species absence/presence than richness.
4. The discussion about Conservation Implications is quite messy, with several distinct ideas mixed within the same paragraph, and without clear statement sentences or concluding sentences. Moreover, there is an extensive literature review, and it is difficult to distinguish which arguments are derived from such review or from the results. I recommend a more straightforward discussion about the conservation implications. For example, given the result that várzea and terra firme forests are interconnected habitats that are seasonally utilized by a suite of mobile species, the authors might organize the discussion in this way (it is just a suggestion): (1) the synergistic effect of having threats from connected terrestrial and folded systems, (2) the importance of protecting such forests in a spatially connected way (and also in management), (3) the current way in which these systems are being protected (pros and cons), and (4) the challenges of protecting such systems.
5. Line 358. This (and others) paragraph presents a mix of different ideas. The statement sentence is about the anthropogenic threats, but the rest of the paragraph is an extensive revision about hunting (who relevance or relation to the results is not clear).
6. Lines 375-376. "The existing protected area network and management policies were created principally to protect terrestrial ecosystems and are therefore not adequate to mitigate the aforementioned impacts". Please clarify which are the aforementioned impacts (only hunting?). Moreover, why protected areas that are designed to protect terrestrial ecosystems cannot mitigate hunting? I think the big problem of Amazonian reserves is more related to the challenge of monitoring remote and big areas, and the limited budget.
7. Line 375. I do not understand the main idea of this paragraph. Is about the current problems of Amazonian protected areas? The authors seem to criticize the establishment of protected areas with sustainable-use because their limited effectiveness. But, what about the importance of such type management for local populations and their livelihood? There is a lot of literature focused on understanding the complex effect of management types. Be careful.
8. Line 389. What is the main conclusion or idea of this paragraph? That varzea and terra firme have different threats and... ? It feels like just a description of threats, without a conclusion or implications.
9. Lines 406-408: "Different Amazonian forest types exhibiting staggered resource pulses must be included within the same or neighboring sustainable-use protected areas". This sounds contradictory, because I got the impression that the authors criticize the protected area with sustainable use. Moreover, creating large areas that include terra firme, varzea, etc has some important challenges / implications.

Annotated reviews are not available for download in order to protect the identity of reviewers who chose to remain anonymous.

---

## Round 0.2 · accepted · Accept

Thanks for taking the time to modify your manuscript based on the reviewer comments, which have helped to improve it for clarity and focus. I note that you will consider providing the imagery in an online repository - thank you.

#